# Potential Non-Invasive Biomarkers for Early Diagnosis of Oral Squamous Cell Carcinoma

**DOI:** 10.3390/jcm10081658

**Published:** 2021-04-13

**Authors:** Valentina Dikova, Eloisa Jantus-Lewintre, Jose Bagan

**Affiliations:** 1Faculty of Medicine and Dentistry, University of Valencia, 46010 Valencia, Spain; v.dikova@yahoo.com; 2Molecular Oncology Laboratory FIHGUV, 46014 Valencia, Spain; jatus_elo@gva.es; 3Mixed Unit TRIAL (CIPF-FIHGUV), 46012 Valencia, Spain; 4Department of Biotechnology, Universitat Politècnica de València, 46022 Valencia, Spain; 5CIBERONC, 46014 Valencia, Spain; 6Service of Stomatology and Maxillofacial Surgery, General University Hospital of Valencia, 46014 Valencia, Spain

**Keywords:** oral cancer, oral leukoplakia, non-invasive biomarkers, saliva testing, translational research

## Abstract

This study aimed to investigate the role of a panel of salivary cytokines as biomarkers for early detection oral squamous cell carcinoma (OSCC), comparing their levels among healthy individuals, patients with oral leukoplakia (OL), and malignant lesions. Cytokine profiling analysis performed in a minimally invasive sample was correlated with clinicopathological variables in our patient cohorts. Unstimulated saliva was obtained from subjects with OSCC at early (*n* = 33) and advanced (*n* = 33) disease, OL with homogeneous (*n* = 33) and proliferative verrucous (*n* = 33) clinical presentations, and healthy controls (*n* = 25). Salivary IL-1α, IL-6, IL-8, IP-10, MCP-1, TNF-α, HCC-1, and PF-4 levels were analyzed by a sensitive bead-based multiplex immunoassay. Mean levels of IL-6, IL-8, TNF-α, HCC-1, MCP-1, and PF-4 differed significantly between OSCC, OL, and control saliva (*p* < 0.05). We found notably higher IL-6 and TNF-α in advanced compared to early OSCC stages. The area under the curve (AUC) for OSCC vs. control was greater than 0.8 for IL-6, IL-8, TNF-α, and HCC-1, and greater than 0.7 for PF-4. The presence of neck metastases (NM) was associated with increased IL-6 and TNF-α levels. Our findings suggest that salivary IL-6, IL-8, TNF-α, HCC-1, and PF-4 may discriminate between OSCC, OL, and healthy controls. IL-6 and TNF-α may indicate OSCC progression, being distinctive in the presence of NM.

## 1. Introduction

Oral squamous cell carcinoma (OSCC) is the most prevalent malignant neoplasia of the oral cavity that accounts for more than 90% of all head and neck (HNC) cancers [1]. It usually affects elderly people over the age of 50. Nevertheless, the occurrence among young adults between 18 and 44 years ranges from 0.4 to 3.6%, with a rising tendency [2]. Oral cancer is a complex multifactorial disease involving personal lifestyle, genetic and environmental factors. Tobacco smoking, betel chewing, excessive alcohol consumption, and exposure to ultraviolet and ionizing radiation, infection with human papillomavirus (HPV), and human immunodeficiency virus (HIV) have long been implicated as major risk factors [2]. The gold standard for diagnosis remains histopathological assessment of an incisional tissue biopsy taken from the suspected area. Specific biomarkers are as yet unavailable and regular tests for the detection of pre-malignant lesions remain scarce in routine clinical settings. Some OSCCs are preceded by clinically visible but otherwise often asymptomatic lesions of the oral mucosa, summarized as precancerous and considered as oral potentially malignant disorders (OPMD) according to the World Health Organization (WHO) [3]. Among them, oral leukoplakia (OL) is the most common representative, considered to have the highest risk of malignant transformation reaching up to 17% [3,4]. The latter exhibits different subtypes out of which the non-homogeneous proliferative verrucous leukoplakia (PVL) presents a higher tendency for cancerous conversion than the homogeneous clinical form according to follow-up studies [5,6]. Immunological analysis of OL specimens revealed the presence of different inflammatory cells in connective tissue, suggesting chronic inflammation [7]. The relation between oral cancer/precancer and chronic inflammation has been proved by the imbalance in local and systemic immunomodulatory cytokine levels that may promote tumor growth and proliferation [7,8]. Cytokines including interleukins (IL), chemokines, interferons (IFNs), and tumor necrosis factors (TNFs) are involved in the regulation of the innate and adaptive immune responses [9,10]. Pro-inflammatory cytokines such as IL-1, IL-6, IL-8, IFN-γ, TNF-α, on one side are responsible for the growth and proliferation of immune and tumor cells, while on the other hand, increase tumor immune surveillance programs. In contrast, anti-inflammatory factors like IL-1 receptor antagonist (IL-1ra) IL-4, IL-10, IL-13, neutralize the proliferative potential of their pro-inflammatory counterparts and, at once, negatively regulate the anti-tumor immune response [11,12]. Functionally classified as inflammatory and homeostatic, chemokines are involved in chemotaxis in inflammation and angiogenesis, as well important members of the tumor microenvironment, mediating the recruitment of immune cells in it [13]. Pathway intercommunications are of major importance, as single or a combination of several cytokines can contribute to up- or down-regulation of others, and certain ones can have both properties. It has been suggested that overexpression of pro-inflammatory cytokines promotes tumor cell growth and survival [13,14]. Literature highlights the altered production of nuclear factor kappa-light-chain-enhancer of activated B cells (NF-κB) dependent modulators such as IL-1, IL-6, IL-8, IL-10, IL-13, and TNF-α by OSCC cells [14,15,16,17]. Besides, the NF-κB family of transcription factors modulates the transcription of angiogenic and tumorigenic chemokine genes, in part. Overall, altered chemokine function in cancer promotes cell survival, enhances proliferation, neovascularization, motility, and metastasis in multiple tumor types [15]. Although most of the investigations have addressed OSCC immune deregulation by using serum as a cytokine source [18,19], sometimes, saliva may be preferred to blood or other body fluids, due to its continuous contact with the oral lesions, offering less invasive collection, processing, and storage methods [20]. In laboratory practices, immunoassays are commonly utilized for the evaluation of cytokine expression because of their specificity and sensitivity [21]. To date, no study has analyzed differences in salivary cytokines between early and advanced OSCC stages, comparing the results with two OL subgroups as potential pre-cancer disorders: one with the lowest risk, being homogeneous leukoplakia (HL), and one with the highest risk of malignant transformation, being PVL. The main purpose of this study was to investigate a panel of salivary cytokines as putative biomarkers to discriminate oral cancer from the two OL subtypes and controls, focusing on discernment between early OSCC and the non-malignant cases analyzed. In addition, we intended to study if any of the target biomarkers could indicate the group of cases with neck metastasis within the cancer patients.

## 2. Materials and Methods

### 2.1. Study Participants and Sample Collection

Samples were obtained at the Service of Stomatology and Maxillofacial Surgery and analyzed in the Molecular Oncology laboratory of the University General Hospital of Valencia (HGUV) between 2017 and 2020. The 157 volunteers enrolled in the study were distributed in five groups after established case-control inclusion criteria. Group 1 and 2 consisted of 33 patients each, of early (Tis, I and II) and advanced (III and IV) OSCC stages, respectively. The diagnosis was based on histopathological analysis from a tissue biopsy and staged according to the TNM classification system [22]. Group 3 was composed of 33 HL and Group 4 of 33 PVL cases. The diagnosis of OL was done according to the Van der Waal criteria [23,24]. Like OSCC, OL patients were not currently undergoing or having undergone any form of treatment for these lesions. The control group (group 5) included 25 age and sex-matched healthy individuals, without visible oral lesions and any acute or chronic inflammatory conditions. Oral health and periodontal status were recorded for all the volunteers recruited in the different groups and those presenting oral significant periodontal disease were not considered for analysis. We could not identify any systemic diseases (such as autoimmune disorders) in our groups that may have influenced our salivary findings. We have not analysed the presence of HPV infection in our cases in the different groups. Therefore, we had no information available on their HPV-status.

All the participants were asked to refrain from eating, drinking, smoking, and using oral hygiene products for at least 1 h prior to sample collection, done by expectoration into 15 mL sterile tubes for 5 min. Unstimulated whole saliva was immediately centrifuged at 3000 rpm for 15 min at 4 °C, clarified supernatant was separated and frozen at −80 °C until further use [25]. Specimens with visible blood traces were discarded from the study.

### 2.2. Multiplex Immunoassay

Samples were thawed on ice and centrifuged at 1500 rpm at 4 °C for 10 min. For cytokine quantification, samples were diluted in assay buffer (1:2) to reduce viscosity. IL-1α, IL-6, IL-8, IP-10, MCP-1, TNF-α, HCC-1, and PF-4 concentrations were determined using MILLIPLEX MAP Human Cytokine/Chemokine assay kits (EMD Millipore, Burlington, MA, USA), as per manufacturer’s protocols. The IL-1α, IL-6, IL-8, IP-10, MCP-1, TNF-α, and HCC-1 standard curves ranged from 0 to 10,000 pg/mL and the lower limits of detection were 9.4 pg/mL, 0.9 pg/mL, 0.4 pg/mL, 8.6 pg/mL, 1.9 pg/mL, and 0.7 pg/mL, and 2.0 pg/mL, respectively. The PF-4 standard curve ranged from 0 to 150 ng/mL with a minimum detectable dose of 0.005 ng/mL. The analysis was carried out on a Luminex^®^ 200 (Austin, TX, USA) instrument, while a digital processor managed data output and XPONENT^®^ software (Luminex^®^ 200) returned data as Median Fluorescence Intensity (MFI), analyzed using a 5-parameter logistic (5 PL) method for calculating cytokine concentrations in pg/mL per each sample. All samples were assayed in duplicate.

### 2.3. Statistical Analysis

The demographic variables were presented as simple descriptive statistics calculating median and standard deviation (SD) of numerical data-like age. Because the cytokine variables did not follow a normal distribution, the comparisons with categorical variables were conducted using the non-parametric Mann–Whitney U-test. *p* values ≤ 0.05 were accepted as statistically significant. The performance of each parameter in the prediction of disease vs. control status was evaluated by means of the receiver operating characteristic (ROC) curve, and the area under the curve (AUC) was measured. Differences between ROC AUCs were estimated according to the DeLong method [26] and binomial exact confidence intervals (CI) for the AUCs were calculated. The best cut-off value was selected using the Youden index [27]. Pearson’s correlation test was carried out to evaluate potential relationships among the cytokines. FDR adjusted *p*-value < 0.05 was considered significant with CI of 95%. The Bayesian hypothesis testing was used to estimate conditional probabilistic relationships between cytokine levels and patients’ clinical phenotypes. Statistical computing was conducted on GraphPad Prism version 6.0 (San Diego, CA, USA), MedCalc version 19.6 (Ostend, Belgium), and R version 4. 0. 2 (The R Foundation, Vienna, Austria).

## 3. Results

### 3.1. Salivary Cytokine Levels in OL, OSCC Patients, and Healthy Controls

The demographic and clinical characteristics of control individuals, HL, PVL, and OSCC patients are shown in Table 1. There were no significant differences in age and sex, among patients and controls (*p* > 0.05). The mean concentrations (pg/mL) of the eight salivary biomarkers are presented in Table 2. We found that patients with HL and PVL have significantly higher salivary IL-6, IL-8, MCP-1, TNF-α, and HCC-1 than their healthy counterparts (Table 2). No important differences were estimated in the expression of the target proteins between the two leukoplakia clinical forms. Further comparisons revealed notable elevations of IL-6, IL-8, TNF-α, HCC-1, and PF-4 levels in OSCC collated to OL sample cohorts (Figure 1), with increases traceable from early oral cancer stages (Table 2). The optimal predictive models based on these cytokines to distinguish OSCC from OL lesions yielded significant AUCs greater than 0.7 (Figure 1), the diagnostic utility of which is described in Table 3. Comparison of individual ROC AUCs showed no significant differences (*p* > 0.05, DeLong et al.) between IL-6, TNF-α, and PF-4, as well as among IL-8, PF-4, and HCC-1, outlining them as evenly suitable biomarker candidates. The multi-marker ROC curve of all five cytokines yielded an AUC of 0.884 (74.24% sensitivity and 80.30% specificity) (Figure 1). We also found the levels of six biomarkers including IL-6, IL-8, MCP-1, TNF-α, HCC-1, and PF-4 considerably increased in OSCC compared to control saliva (Figure 2), with elevation detectable from early disease onset (Table 2). No appreciable differences were assessed in IL-1α and IP-10 levels among the collated groups. ROC curve analysis showed AUCs greater than 0.8 for IL-6, IL-8, TNF-α, and HCC-1, and 0.710 for PF-4 (Figure 2). The potential diagnostic utility of those factors is presented in Table 3B. No significant differences were found between IL-6, TNF-α, and HCC-1 AUCs, as well as between IL-6, IL-8, and HCC-1 suggesting them as equally valuable biomarker candidates. The cut-off value for IL-6 that best distinguishes OSCC patients from controls was 15.038 pg/mL (81.82% sensitivity and 96% specificity), for IL-8 was 923.78 pg/mL (75.76%sensitivity and92% specificity), for TNF-α was 17.01 pg/mL (86.36% sensitivity and 100% specificity), and for HCC-1 was 112.90 pg/mL (83.33% sensitivity and 92% specificity) selected by the Youden index. The optimal predictive model based on the six cytokines together, to classify OSCC and control subjects, yielded an AUC of 0.980 (95.45% sensitivity and 88% specificity) (Figure 2).

In addition, IL-6 and TNF-α levels marked considerable growth in patients at advanced compared to early OSCC clinical stages (Table 2) with AUCs of 0.694 (95%CI: 0.595–0.802) and 0.684 (95%CI: 0.558–0.793), respectively.

### 3.2. Correlation among Salivary Cytokine Levels and Association with Clinicopathological Characteristics of OSCC Patients

To find potential correlations among cytokine levels within the control and OSCC groups, Pearson’s correlation test was performed. In the control cohort (Table 4), IL-1α was positively correlated with IL-8 and TNF-α. IL-6 showed a positive association with IL-8, MCP-1, TNF-α, and HCC-1. IL-8 with HCC-1 and TNF-α. A positive relationship was also found between TNF-α and HCC-1, as well as among HCC-1 and PF-4. Across the early OSCC group (Table 4), positively correlated were IL-1α with MCP-1, and TNF-α, also IL-6 with IL-8 and TNF-α. As well as IL-8 with MCP-1, TNF-α, HCC-1, and PF-4. MCP-1 was positively associated with TNF-α, and the latter with HCC-1. Within the advanced OSCC group (Table 4), IL-6 was positively correlated with IP-10 and TNF-α. IL-8 with MCP-1, HCC-1, and PF-4. Additionally, MCP-1 with HCC-1 and PF-4. All of the aforementioned correlations were statistically significant (*p* ≤ 0.05). Besides, logistic regression analysis revealed a significant correlation of TNF-α and IL-6 with advanced OSCC (*p* ≤ 0.001 and 0.008, respectively). Bayesian statistics based on the interpretation of conditional probability were utilized to find potential associations between cytokine levels and OSCC patients’ clinical characteristics. A direct relationship between OSCC, IL-6, and TNF-α expressions was observed with the likelihood to indicate advanced disease higher than 70% (*p* = 0.76, 0.77, respectively) (Table 5). Besides, the probability of association between IL-6 and the presence of neck metastases exceeded 80% (*p* = 0.81) (Table 5). No significant correlation was found among patients’ sex, smoking habits, lesion/tumor location, clinical form, and cytokine expression.

## 4. Discussion

The lack of symptoms at initial OSCC stages often results in late diagnosis, emphasizing the need for a practical and simple method for early detection to be used for definitive diagnosis, as well as for screening programs. Since inflammation has been linked to the pathogenesis of OSCC, the research to date indicates the possibility of using pro- and anti-inflammatory factors as screening tools for those patients [16,17]. It is now well recognized that altered cytokine responsiveness is tightly related to the development of oral cancer. Besides, it has also been associated with premalignant lesions such as OL [28,29]. The multiplex cytokine test was efficient in the detection and quantification of cytokine levels in the saliva of patients with HL PVL, at different clinical stages of OSCC, and their healthy counterparts. Six biomarkers with significantly different expression in OSCC than in controls were found in our study: IL-6, IL-8, TNF-α, MCP-1, HCC-1, and PF-4 (Figure 2), being discriminately increased from early disease stages. Among them, IL-6 and TNF-α marked a considerable growth towards the OSCC evolution, indicating potential involvement in disease progression and severity. Our results are consistent with previous findings where increased levels of NF-κB associated IL-6, IL-8, and TNF-α in oral cancer saliva have been reported, suggesting that OSCC progression is likely enhanced by continued expression of pro-inflammatory and pro-angiogenic cytokines [30,31,32,33]. However, there is a scarcity of studies describing cytokine responsiveness at different OSCC clinical stages. Lee et al. [33] found significantly upregulated IL-6, IL-8, and TNF-α levels in early OSCC (I+II stages) compared to control subjects, but no distinction between early and advanced disease (III+IV stages). Similarly, Dineshkumar et al. [34] stated no significant difference in salivary IL-6 based on OSCC clinical staging. Krishnan et al. [35] revealed important TNF-α overexpression in the sputum of OSCC patients at stage IV compared to clinical stages I, II, and III. According to our estimations, no differential IL-1α levels were seen between controls, early and advanced OSCC, corroborating the findings of Lee et al. [33] and Babiuch et al. [29]. Chemokines are secreted in response to signals such as pro-inflammatory members of the IL-1 family, TNFs, and interferon-c (IFN-c), and thus playing an important role in selectively recruiting monocytes, neutrophils, and lymphocytes [36]. Besides, their functions are multifaceted including inflammation and/or immune response [37]. The complex relationship among these immune modulators in OSCC was demonstrated by Pearson’s correlation analysis displaying multiple significant positive interconnections between the investigated markers (Table 4). Studies have implicated several chemokines and their receptors in squamous cell cancers of the HNC, arguing that tumor-related changes in chemokine composition are detectable in oral fluid [38,39]. Indeed, our results showed considerable overexpression of inflammatory MCP-1 along with homeostatic HCC-1 and PF-4 in the saliva of patients at early OSCC stages compared to normal controls (Table 2). Monocyte chemoattractant protein 1 (MCP-1) or CCL2, regulates monocyte migration and is frequently expressed as tumor cell-associated chemokine [40]. Increased MCP-1 in OSCC with metastatic lymph nodes has been detected by Ferreira and co-workers [41] while its expression in HNC has been associated with tumor invasion in esophageal SCC [42]. HCC-1 or hemofiltrate C-C motif chemokine 14 (CCL14) is a homeostatic chemokine found to promote angiogenesis and tumor progression [43]. Its involvement in oral carcinogenesis has been reported by Feng et al. [44] exhibiting differentially expressed long non-coding RNA (lncRNA) HCC-1 transcript in oral mucosal samples from OSCC patients. Platelet factor 4 (PF-4) or CXCL14, also known as BRAK is a highly conserved homeostatic chemokine, responsible for immune cell recruitment, maturation, and its influence on epithelial cell motility is thought to be a key modulatory factor in cancer. Dysregulated PF-4 was shown to limit critical antitumor immune regulation and to correlate to poor patient prognosis [45]. Notably increased PF-4 concentration in OSCC saliva was firstly described in the current research. In contrast, decreased PF-4 expression has been found in OSCC cells and its induced up-regulation resulted in suppressed activity toward oral cancer tumor progression in vivo [46,47].

The present outcomes revealed higher IP-10 levels in OSCC than in control saliva, yet, this was not statistically significant. Elevated IP-10 levels have previously been detected in serum from HNSCC patients [48], as well as in tissue and peripheral blood of individuals with nasopharyngeal carcinoma [49]. Besides, it has been suggested as a potential marker for radiotherapy response and overall survival in patients with tongue SCC [50].

It is now well known that many OSCC cases have arisen from preceding oral potentially malignant disorders (OPMDs), such as leucoplakia [51,52]. Although cytokine modulations in those patients have been researched in diverse samples, there is a lack of studies considering more than one cytokine when comparing distinct leukoplakia manifestation forms. To the best of our knowledge, we firstly described considerable elevation of IL-6, IL-8, TNF-α, IP-10, HCC-1, and PF-4 levels in OSCC compared to OL patients (Figure 1), being detectable from early cancer stages. Besides, all of the listed analytes, except PF-4, were significantly higher in HL and PVL than in subjects without oral lesions (controls) (Table 2). Whereas the premalignant microenvironment suggests eliciting pro-inflammatory cytokine production, the tumor microenvironment seems more immune-stimulatory. It could be assumed that malignant transformation of HL and PVL may be influenced by the continuous exposure to the overexpression of these pro-inflammatory, proangiogenic mediators. The fact that the same molecules were remarkably elevated in OSCC and OL may have a prognostic significance for the malignant potential of leukoplakia lesions. A few studies have addressed the potential of routine cytokine measurements in OSCC screening. Rhodus et al. [15] reported significantly higher salivary levels of IL-6, IL-8, and TNF-α in patients with OSCC compared to ones with oral preneoplastic lesions. In accordance, other authors observed elevated IL-6 [34], IL-8 [53,54], and TNF- α [35] in OSCC saliva compared to OL and normal controls. Salivary MCP-1, IP-10, HCC-1, and PF-4 expressions in OL patients have been characterized and compared to the ones in OSCC, primarily in this study. The significant increase marked by HCC-1 and PF-4 in OSCC, compared to OL, being detectable from the early onset of this malignancy brings new insights into the identification of prognostic, liquid biopsy-derived candidate biomarkers for oral cancer.

The discriminatory efficacy of the studied markers to discern OSCC from OL and individuals without oral lesions (controls) we assessed by the means of ROC curve analysis. According to our data, IL-6, TNF-α, and PF-4 demonstrated the highest potential in discrimination of OSCC from OL patients, while IL-6, TNF-a, IL-8, and HCC-1 appeared effective in classifying OSCC and controls. Superior sensitivity of IL-6 and IL-8 in oral cancer detection has also been evaluated in two large-scale studies [7] and [34], respectively. Furthermore, a combined biomarker panel demonstrated improved sensitivity and specificity for distinguishing between patients and controls. This foundation suggests that a multi-marker signature can yield satisfactory accuracy for the detection of OSCC.

The conducted correlation analysis revealed no association between altered cytokine responsiveness and OL or OSCC patient demographics and clinical features such as sex, smoking habits, lesion location, and clinical form. We assume that the sample cohorts were not large enough to confirm the statistical significance of potential relationships. However, an association was established between advanced OSCC and increased IL-6 and TNF-α levels. Their gradual, significant growth put forward involvement in accelerating disease progression, consistent with the findings of Jablonska et al. [55]. We also found a relationship between increased IL-6 concentration and the presence of neck metastases in oral cancer patients, which is in line with studies by Riedel et al. [56] and Tartour et al. [57] revealing a significant association between positive lymph nodes in HNSCC and elevated serum IL-6 levels.

NF-κB-dependent cytokines are molecular messengers highly involved in inflammation, angiogenesis, and proliferation [7,16,29], the altered expression of which have also been reported in non-malignant oral pathologies [15,29]. Abundant expression of cytokines and chemokines has been detected in the course of periodontitis [58,59]. To avoid any potential interference, our cohorts of volunteer donors had been pre-screened to eliminate those with any local acute or chronic inflammation (including periodontal disease) in the oral cavity upon direct clinical visual exploration. With regards to cancer risk, cytokines are of particular interest as they are involved in periodontal pathogenesis but also expressed in healthy sites [60]. Nevertheless, considering that local inflammatory conditions may result in cytokine overexpression, investigators have found the contribution of OSCC to the elevation of these modulators to outweigh any potential background conferred by the host’s inflammatory condition [59,60].

## 5. Conclusions

Our comparative profiling analysis suggests that saliva-derived IL-6, IL-8, TNF-α, HCC-1, and PF-4 may discriminate between OSCC, OL patients, and healthy controls. These non-invasive biomarkers may serve a useful role in early disease detection, as well as for screening of patients at risk of developing oral cancer. The considerable growth of TNF-α and IL-6 concentrations towards OSCC evolution, and IL-6 being distinctive in the presence of neck metastasis suggests their potential involvement in disease progression and severity. Putative biomarkers used in combination may enhance their accuracy, so a multi-marker signature for prognosis and diagnosis needs to be elaborated to yield satisfactory precision.

## Figures and Tables

**Figure 1 jcm-10-01658-f001:**
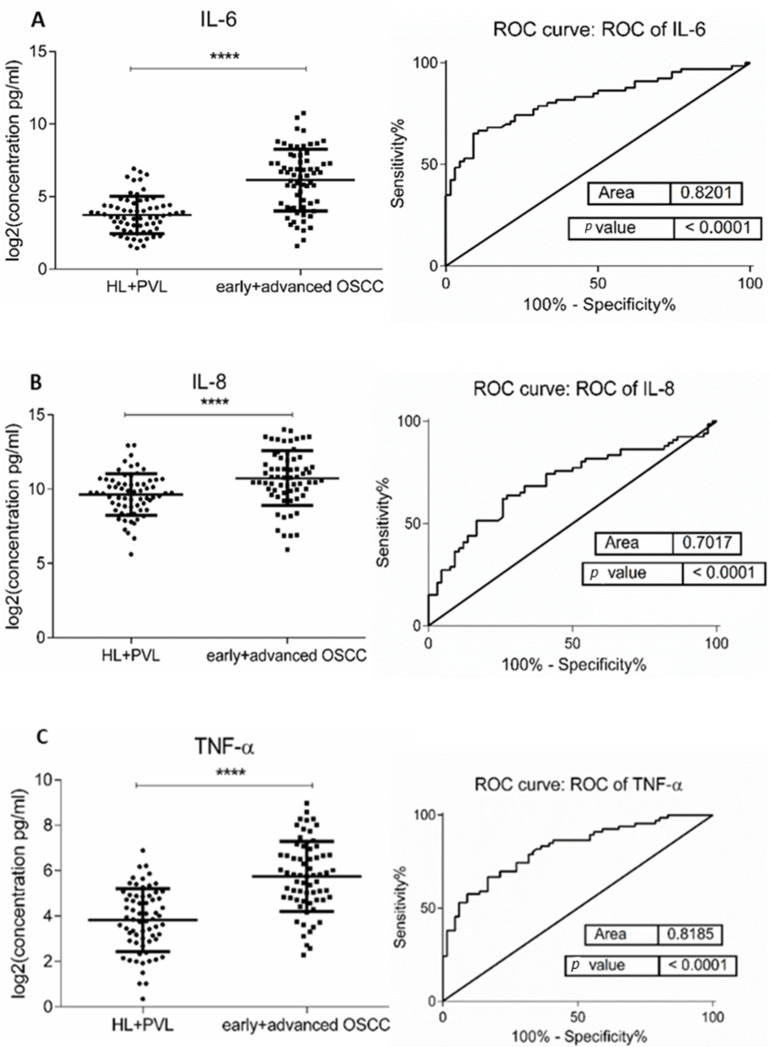
Comparison of salivary cytokine levels between OL (HL + PVL) and OSCC (early +advanced stages) patients. Dot plot of significantly different log2 levels (pg/mL) (left) and ROC curve (right) of (**A**) IL-6, (**B**) IL-8, (**C**) TNF-α, (**D**) HCC-1, and (**E**) PF-4 (**** *p* ≤ 0.0001, 0.0001, 0.0001, 0.0001, and 0.0001, respectively). (**F**) ROC curve combining multiple markers including IL-6, IL-8, TNF-α, HCC-1, and PF-4. Values represent mean ± SD of *n* = 66 for OL and OSCC groups, where *n* is an average of two technical replicates.

**Figure 2 jcm-10-01658-f002:**
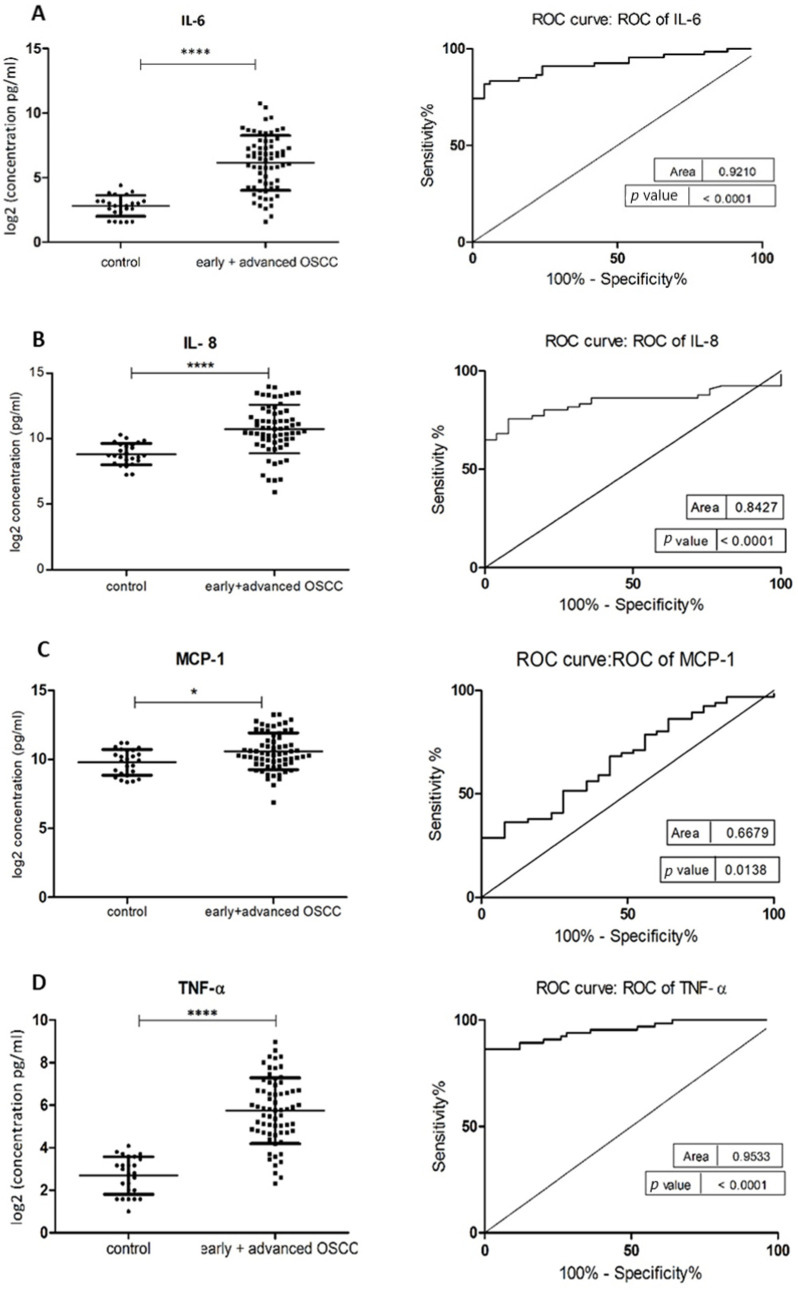
Comparison of salivary cytokine levels between control individuals and OSCC patients. Dot plot of significantly different log2 levels (pg/mL) (left) and ROC curve (right) of (**A**) IL-6, (**B**) IL-8, (**C**) MCP-1, (**D**) TNF-α, (**E**) HCC-1 and (**F**) PF-4 (*p* ≤ 0.0001 ****, 0.0001 ****, 0.01 *, 0.0001 ****, 0.0001 **** and 0.002 **, respectively). (**G**) ROC curve combining multiple markers, including IL-6, IL-8, TNF-α, MCP-1, HCC-1, and PF-4. Values represent mean ± SD of *n* = 25 (control) and *n* = 66 (early + advanced OSCC stages), where *n* is an average of two technical replicates.

**Table 1 jcm-10-01658-t001:** Demographic characteristics of control subjects (*n* = 25), homogeneous leukoplakia (HL) (*n* = 33), proliferative verrucous leukoplakia (PVL) (*n* = 33) and oral squamous cell carcinoma (OSCC) patients at early (*n* = 33) and advanced (*n* = 33) stages.

				Early	Advanced
	Control	HL	PVL	OSCC	OSCC
Age (median ± SD)	62 ± 8.3	68 ± 12.1	67 ± 12.3	73 ± 10.9	65 ± 15.6
Sex					
Male	9 (36%)	10 (30.3%)	13 (39.4%)	13 (39.4%)	20 (60.6%)
Female	16 (64%)	23 (69.7%)	20 (60.6%)	20 (60.6%)	13 (39.4%)
Tobacco smoking status					
Non-smokers	21 (84%)	15 (45.5%)	24 (72.7%)	23 (69.7%)	14 (42.4%)
Current smokers	4 (16%)	18 (54.5%)	9 (27.3%)	10 (30.3%)	19 (57.6%)
Clinical form					
Homogeneous white plaque/s		33 (100%)			
Verrucous			14 (42.4%)		
Homogeneous and verrucous regions			19 (57.6%)		
Erythroplakic			-	6 (18.7%)	1 (3%)
Exophytic				7 (21.8%)	1 (3%)
Ulcerative				14 (43.7%)	25 (75.7%)
Mixed				5 (15.6%)	6 (18.2%)
Histologic tumor differentiation					
well				26 (78.8%)	19 (57.6%)
not well *				7 (21.2%)	14 (42.4%)
Neck metastases					
yes				6 (18.2%)	26 (78.8%)
no				27 (81.8%)	7 (21.2%)

* Includes moderately and poorly differentiated tumors; HL—homogeneous leukoplakia; PVL—proliferative verrucous leukoplakia; OSCC—oral squamous cell carcinoma.

**Table 2 jcm-10-01658-t002:** Cytokine concentrations (pg/mL) in saliva samples of (**A**) control subjects, patients with HL and PVL, and (**B**) at early and advanced OSCC stages; values are expressed as the arithmetic mean ± standard error of the mean (SEM).

A	Control (*n* = 25)	HL (*n* = 33)	PVL (*n* = 33)		*p* ^a^
IL-1α	1227.24 ± 122.90	1768.11 ± 376.97	1886.48 ± 294.87		ns
IL-6	7.95 ± 0.95	22.61 ± 4.78	18.90 ± 3.75		0.001 **
IL-8	526.17 ± 59.03	1382.92 ± 279.60	1140.87 ± 240.48		0.004 **
IP-10	884.97 ± 93.39	1649.98 ± 386.48	1157.80 ± 251.30		ns
MCP-1	1066.61 ± 126.54	2535.35 ± 372.84	3600.83 ± 481.32		0.001 **
TNF-α	7.62 ± 0.84	19.65 ± 2.82	23.08 ± 4.26		0.001 **
HCC-1	75.36 ± 6.10	174.55 ± 27.66	189.28 ± 35.77		0.002 **
PF-4	253.74 ± 23.98	293.92 ± 60.53	454.78 ± 146.61		ns
**B**	**Early OSCC (*n* = 33)**	**Advanced OSCC (*n* = 33)**	***p*^b^**	***p*^c^**	***p*^d^**
IL-1α	1556.19 ± 248.31	1313.79 ± 189.58	ns	ns	ns
IL-6	99.82 ± 26.09	262.08 ± 65.83	<0.001 **	<0.001 **	0.01 **
IL-8	2567.01 ± 549.72	4124.81 ± 787.23	<0.001 **	0.05 *	ns
IP-10	1963.89 ± 365.41	1556.53 ± 332.42	ns	ns	ns
MCP-1	2560.73 ± 370.26	2099.76 ± 429.95	0.003 **	ns	ns
TNF-α	59.82 ± 11.56	122.52 ± 20.68	<0.001 **	<0.001 **	0.01 **
HCC-1	298.53 ± 49.55	551.32 ± 119.67	<0.001 **	0.01 **	ns
PF-4	642 ± 104.85	1021.85 ± 220.02	0.01 **	<0.001 **	ns

IL, interleukin; IP, interferon gamma-induced protein; MCP, monocyte chemoattractant protein; TNF-α, tumor necrosis factor-alpha; HCC-1, hemofiltrate CC chemokine 1; PF-4, platelet factor 4; HL, homogeneous leukoplakia; PVL, proliferative verrucous leukoplakia; OSCC, oral squamous cell carcinoma. Mann–Whitney U-test for **^a^** control vs. OL (HL + PVL), **^b^** control vs. early OSCC, **^c^** OL vs. early OSCC, and **^d^** early vs. advanced OSCC; * *p* ≤ 0.05; ** *p* ≤ 0.01; ns—non-significant.

**Table 3 jcm-10-01658-t003:** Diagnostic utility of salivary cytokines to distinguish (**A**) OSCC from OL patients and (**B**) OSCC from control individuals.

A Diagnostic Parameters	IL-6	IL-8	TNF-α	HCC-1	PF-4
AUC	0.820	0.702	0.819	0.714	0.731
*p*	<0.0001	<0.0001	<0.0001	<0.0001	<0.0001
SE ^a^	0.037	0.046	0.036	0.045	0.043
YI	0.560	0.363	0.500	0.363	0.348
Cut-off value (pg/mL)	>38.314	>1281.38	>34.08	>159.19	>129
Sensitivity %	66.67	63.64	66.67	71.21	92.42
Specificity %	89.39	72.73	83.33	65.15	42.42
95% CI ^b^	0.744–0.882	0.616–0.778	0.742–0.880	0.629–0.789	0.647–0.804
**B** **Diagnostic Parameters**	**IL-6**	**IL-8**	**TNF-α**	**HCC-1**	**PF-4**
AUC	0.921	0.842	0.953	0.898	0.710
*p*	<0.0001	<0.0001	<0.0001	<0.0001	0.002
SE ^a^	0.026	0.040	0.019	0.032	0.053
YI	0.778	0.677	0.863	0.753	0.436
Cut-off value (pg/mL)	>15.038	>923.78	>17.01	>112.909	>308
Sensitivity %	81.82	75.76	86.36	83.33	63.64
Specificity %	96.00	92.00	100	92.00	80.00
95% CI ^b^	0.849–0.969	0.616–0.778	0.886–0.986	0.886–0.9	0.605–0.800

AUC—area under the curve; *p*—significance level P (Area = 0.5); SE- standard error ^a^ DeLong et al. [26]; YI—Youden index (J), CI—confidence interval ^b^—Binomial exact.

**Table 4 jcm-10-01658-t004:** Pearson’s pairwise correlation among salivary cytokines in (**A**) control, (**B**) early OSCC and (**C**) advanced OSCC groups. Correlation is significant when *p* ≤ 0.05 (*), *p* ≤ 0.01 (**) and *p* ≤ 0.001 (***).

A			Control			
		IL-8	MCP-1	TNF-α	HCC-1	PF-4
IL-α	Pearson correlation	* 0.426		* 0.458		
	Sig. (two-tailed)	0.03		0.02		
IL-6	Pearson correlation	** 0.593	* 0.493	*** 0.817	* 0.438	
	Sig. (two-tailed)	0.002	0.01	<0.001	0.03	
IL-8	Pearson correlation			*** 0.706	* 0.512	
	Sig. (two-tailed)			<0.001	0.01	
TNF-α	Pearson correlation				* 0.463	
	Sig. (two-tailed)				0.02	
HCC-1	Pearson correlation					** 0.601
	Sig. (two-tailed)					0.001
**B**			**Early OSCC Stages**			
		**IL-8**	**MCP-1**	**TNF-α**	**HCC-1**	**PF-4**
IL-1α	Pearson correlation		* 0.364	** 0.528		
	Sig. (two-tailed)		0.04	0.002		
IL-6	Pearson correlation	** 0.478		** 0.456		
	Sig. (two-tailed)	0.005		0.008		
IL-8	Pearson correlation		** 0.515	*** 0.619	** 0.522	* 0.350
	Sig. (two-tailed)		0.002	<0.001	0.002	0.05
MCP-1	Pearson correlation			* 0.393		
	Sig. (two-tailed)			0.02		
TNF-α	Pearson correlation				*** 0.595	
	Sig. (two-tailed)				<0.001	
**C**			**Advanced OSCC Stages**			
		**IP-10**	**MCP-1**	**TNF-α**	**HCC-1**	**PF-4**
IL-6	Pearson correlation	** 0.457		*** 0.568		
	Sig. (two-tailed)	0.007		<0.001		
IL-8	Pearson correlation		*** 0.549		*** 0.558	*** 0.738
	Sig. (two-tailed)		<0.001		<0.001	<0.001
MCP-1	Pearson correlation				*** 0.569	** 0.495
	Sig. (two-tailed)				<0.001	0.003
HCC-1	Pearson correlation					* 0.369
	Sig. (two-tailed)					0.03

**Table 5 jcm-10-01658-t005:** Conditional concentration ranges of salivary cytokines (pg/mL) that maximize the probability of association with (**A**) advanced OSCC and (**B**) the presence of neck metastases in OSCC patients (early + advanced stages).

A Advanced OSCC Stages					
	Mean	L	U	P (Advanced OSCC)	P (Early OSCC)
IL-1α	1703.27	1690.87	1715.66	0.512 (±0.07)	0.483 (±0.07)
IL-6	324.00	319.08	328.92	0.763 (±0.06)	0.235 (±0.07)
IP-10	2195.84	2175.20	2216.47	0.555 (±0.07)	0.455 (±0.06)
MCP-1	2674.38	2654.15	2694.60	0.486 (±0.06)	0.53 (±0.06)
TNF-α	156.37	155.13	157.60	0.772 (±0.06)	0.228 (±0.07)
HCC-1	573.70	568.87	578.52	0.575 (±0.06)	0.422 (±0.06)
PF-4	994.85	986.26	1003.43	0.529 (±0.07)	0.456 (±0.07)
**B Neck Metastases (NM)**					
	**Mean**	**L**	**U**	**P (Present NM)**	**P (Absent NM)**
IL-1α	1706.22	1694.38	1718.06	0.582 (±0.06)	0.421 (±0.06)
IL-6	313.45	308.94	317.97	0.817 (±0.06)	0.181 (±0.06)
IP-10	2171.82	2152.56	2191.08	0.606 (±0.06)	0.387 (±0.07)
MCP-1	2884.44	2864.18	2904.69	0.569 (±0.07)	0.423 (±0.06)
TNF-α	136.34	135.25	137.43	0.732 (±0.07)	0.275 (±0.07)
HCC-1	559.69	555.16	564.22	0.634 (±0.06)	0.355 (±0.06)
PF-4	1024.75	1016.33	1033.18	0.607 (±0.06)	0.384 (±0.06)

Mean—average value (pg/mL); L—lower and U—upper value; P—probability of association.

## Data Availability

Not applicable.

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
