# Peer review of "Potential Non-Invasive Biomarkers for Early Diagnosis of Oral Squamous Cell Carcinoma"

_jcm, 2021, doi:10.3390/jcm10081658_

Round 1
Reviewer 1 Report
Thank you for asking me to review this interesting paper "Stage 1: Potential non-invasive biomarkers for early diagnosis of oral squamous cell carcinoma".
I am not sure why the paper starts with the words "Stage 1:". I suggest that this be removed if the paper is to proceed to publication.
The findings in this paper are interesting as they discuss the cytokine levels expressed in saliva in patients with oral leukoplakia compared to those with oral squamous cell carcinoma and health individuals. Other than including a leukoplakia group, this is not overly novel, although required to understand this disease process. My main concern about this paper however is the lack of standardization and control for localised diseases which may induce salivary cytokines such as periodontal diseases and inflammatory mucosal diseases other than leukoplakia. Additionally and importantly, there is a lack of standardisation for systemic conditions which may induce strong cytokine levels and which may also be detected in saliva. These two very important considerations must be addressed in a re-written paper or included as strong limitations and the conclusion re-written, if the paper is to be acceptable for publication.
Furthermore more information on the medical status of patients is required in order to allow readers to interpret the findings without confounding factors. Essentially my concern is that the cohorts are not matched for local and systemic disease presentations. This is a significant limitations and design issue that needs to be addressed.
The notion that the levels of these cytokines indicate that progression to oral cancer is very strong given that these cohorts were not followed to measure progression. What the authors have shown is a distinct level of certain cytokines in different groups of patients with different pathologies at any one time. This does not indicate progression. This must be addressed.
Spelling of words in Table 1 must be fixed for correct English such as Erythroplakic and Exophytic.
The limitations of this work must be discussed.
The conclusions are very strongly worded given the design limitations of the study. These must be reconsidered and limited if the paper is to be accepted.
Author Response
Review Report 1
Stage 1 involves initial manuscript submission and review and as it is indicated in the author's guidelines for manuscript preparation: The title of the paper should include the words “Stage 1”. It is not part of the title of the manuscript and will be removed for further processing.
English revision and changes were performed and are traceable using the "Track Changes" function in the Microsoft Word format of the manuscript.
According to our case-control inclusion criteria, we aimed to select volunteers who do not present any acute or chronic inflammatory conditions upon a direct clinical exploration of the oral cavity. Oral health (Restorative care of the teeth, including fillings, and tooth decay), and periodontal status (Bleeding index, plaque index, gingival index, and probing depth) were recorded for all the patients in the different groups of the study in order to make sure that they had similar oral health status. Those presenting periodontal inflammation were not considered for analysis. This has been added into the Materials and Method section of the manuscript (lines: 102-104). Besides, no statistical differences were estimated in age, sex among OL, OSCC patients, and controls.
The patients’ medical status has not been taken into consideration as systemic diseases are not considered to be significant cofounding factors with an impact on the etiology of OL and OSCC. (Warnakulasuriya S, Kujan O, Aguirre-Urizar JM, Bagan JV, González-Moles MÁ, Kerr AR, Lodi G, Mello FW, Monteiro L, Ogden GR, Sloan P, Johnson NW. Oral potentially malignant disorders: A consensus report from an international seminar on nomenclature and classification, convened by the WHO Collaborating Centre for Oral Cancer. Oral Dis. 2020 Oct 31. doi: 10.1111/odi.13704. Epub ahead of print. PMID: 33128420.)The advanced OSCC group included subjects with oral tumors at clinical stages III and IV, generally considered as patients at progressed disease. The significantly different expression levels of cytokines including IL-6, IL-8, TNF-α, HCC-1, and PF4 were found to follow a trend towards an increase from controls to OL and finally to OSCC. OSCC is a multistage process where the initial presence of precursor lesion evolving into a tumor outgrowth due to uncontrolled transformation from hyperplastic to dysplastic areas (typical for OL) to carcinoma in situ and finally to invasive carcinoma is well established. We assume that due to the discriminately increased concentrations of the suggested cytokines, they may serve a useful role as a complementary tool for screening patients presenting OL lesions to estimate possible malignant potential.
Table 1. Erythroplakic and Exophytic spelling revised.
The conclusions were re-written (lines: 353-361).
Reviewer 2 Report
Dikova et al, used IL-1a, IL-8, IP-10,MCP-1, TNF-a, HCC-1, PF-4 biomarkers to discriminate between OSCC, OL and healthy subjects.
The methods of samples preparations and analysis are standardized in many other previous reports, and the current report may just be an additional confirmation of the same results of different populations.
Some issues in the manuscript need to be addresses:
1- The study did not consider the clinical status of the oral condition and the authors claim that “the contribution of OSCC to the elevation of these modulators to outweigh any potential background conferred by the host’s inflammatory condition”. However, previous studies that have taken this factor into account and found out that between (IL-6 and IL-8), only IL-6 could be a potential marker for OSCC due to the elevated expression level of IL-8 in another oral inflammatory condition, lichen planus. Excluding this aspect in this study undermines the sensitivity of using these cytokines as diagnostic markers (https://doi.org/10.1902/jop.2013.130320).
2- There is a wealth of studies that have analyzed the use of the same cytokines as salivary biomarkers in both premalignant and malignant conditions, the discussion of the current paper needs to highlight and justify any new findings.
3- The authors stated in the discussion the promise of using “pro and anti- inflammatory factors as screening tools”. However, they only focus on the pro-inflammatory aspect of the salivary cytokine profile. Including both pro and anti-tumour cytokines would have given a better overview on the cytokine profile of these patients and the best biomarker candidates.
Author Response
Review Report 2
- The clinical state of the oral condition in each study participant was recorded as we have described answering to Reviewer Report 1.
Status was taken into consideration, and patients presenting any acute or chronic inflammatory conditions including periodontitis, were discarded from the study, as commented in the Discussion section (lines: 344-346) of the manuscript.
The modulation of NF-κB-dependent cytokine levels has been studied in oral premalignant lesions and suggested as potential diagnostic tools for detecting oral cancer. Notably elevated expression levels of IL-8, but also of IL-6, and TNF-α have previously been reported in oral leukoplakia, oral lichen planus, and oral submucosa fibrosis, compared to normal healthy subjects (doi:10.5125/jkaoms.2015.41.4.171; doi:10.1016/j.cyto.2007.11.004). Chronic inflammation is not considered to contribute to the development of all those precancerous lesions, as has been recently published (Warnakulasuriya S, Kujan O, Aguirre-Urizar JM, Bagan JV, González-Moles MÁ, Kerr AR, Lodi G, Mello FW, Monteiro L, Ogden GR, Sloan P, Johnson NW. Oral potentially malignant disorders: A consensus report from an international seminar on nomenclature and classification, convened by the WHO Collaborating Centre for Oral Cancer. Oral Dis. 2020 Oct 31. doi: 10.1111/odi.13704. Epub ahead of print. PMID: 33128420
Although chronic inflammatory conditions have scarcely been associated with the pathogenesis of OSCC (Almangush A, Mäkitie AA, Triantafyllou A, de Bree R, Strojan P, Rinaldo A, Hernandez-Prera JC, Suárez C, Kowalski LP, Ferlito A, Leivo I. Staging and grading of oral squamous cell carcinoma: An update. Oral Oncol. 2020 Aug;107:104799.), and several studies have investigated the expression profiles of salivary IL-6, IL-8, and TNF-α among healthy subjects, patients with oral leukoplakia, and/or other OPMDs, and oral cancer to estimate potential differences among these groups. Our study is in a line with previous findings of increased expression levels as of IL-8. Besides, increased abundance of the same three cytokines has been reported in patients with periodontitis. However, the investigators have found the contribution of this malignancy to the cytokine overexpression based on their notably different concentrations in OSCC compared to OL and periodontitis saliva (doi: 10.4081/oncol.2020.465; doi: 10.3389/fimmu.2015.00214 doi: 10.1111/j.1600-0722.2009.00644.x).
- The majority of the existing oral cancer-related studies report differential cytokine expressions based on comparisons of the mean levels of distinct analytes between OSCC patients and their healthy counterparts without oral lesions (controls). Limited investigations have been conducted on the modulation of salivary cytokines and chemokines in patients with oral cancer, premalignant lesions, especially involving Proliferative Verrucous Leukoplakia, the last considered to have a greater malignant potential than other precancerous lesions (doi: 10.1016/j.oraloncology.2011.05.008; doi: 10.1007/s00784-019-03059-9) and their healthy counterparts. Usually, altered salivary levels have been described based on data from individual measurements with the conventional ELISA immune-based assay. Our data is derived from real-time detection and quantification of multiple analytes in individual saliva samples employing the xMAP Luminex technology providing increased detection range, sensitivity, reproducibility. Our study is the first of its kind to estimate the salivary concentrations of 8 cytokines in OSCC patients at different clinical stages and comparing the results with patients presenting different OL subtypes, highlighted in the Introduction section (lines: 79-83). We firstly described considerable growth of IL-6, IL-8, TNF-α, IP-10, HCC-1, and PF-4 levels in OSCC compared to OL patients (including two leucoplakia subtypes), being detectable from early cancer stages, highlighted in the Discussion section (lines: 299-301). Besides, we primarily investigated and characterized salivary MCP-1, IP-10, HCC-1, and PF-4 in preneoplasic lesions, with the last two chemokines resulting notably increased in OSCC compared to OL giving new insights into the discovery of candidate biomarkers with prognostic means, highlighted in the Discussion section (lines:314-318).
- As our investigation has not focused on the role of immunosuppressive cytokines as biomarker targets, hence a description of such representatives was omitted in the discussion section. However, the literature reveals that both, pro and anti-inflammatory factors act as immunomodulatory agents when it comes to tumor immune responses and may play a useful role in studying of oral disorders, including cancer. A report from 2015 describing the prognostic and diagnostic potential of anti-inflammatory immunosuppressive cytokines (IL-4, IL-10, IL-13, IL-RA) as salivary biomarkers for OSCC has been cited in the Discussion section (line: 241). doi: 10.3109/07357907.2015.1041642.
Reviewer 3 Report
This study aimed to investigate the role of a panel of salivary cytokines as biomarkers to aid in the early detection of oral cavity squamous cell carcinoma. They used a small cohort of study participants grouped into either healthy, OL (2 groups; homogenous and proliferative verrucous), and oral cavity squamous cell carcinoma (2 groups; early and advanced stage). This study concludes that saliva-derived IL-6, IL-8, TNF-a, HCC-1, and PF-4 may be used to differentiate between those with oral cancer, and those with the "premalignant lesion" (OL), as well as healthy controls. Furthermore, it suggests the use of IL-6 and TNF as potential predictive markers in neck metastasis. This reviewer recognizes the importance of this study in contributing to the collective knowledge of this area of research, but numerous studies have already evaluated these cytokine/chemokine biomarkers. For that reason, this reviewer fails to recognize this studies originality and scientific novelty since there are numerous studies and reviews that have already looked at these biomarkers with similar conclusions. There are also significant issues with the manuscript that leads this reviewer to state that it is not acceptable for publication in its current form.
MAJOR issues:
1. The analysis in Table 5 does not include any p-values (significance testing)? was this statistical test not performed? Also, in these types of analyses, one would argue that a univariate and multivariate analysis should be performed, where ALL clinical and biochemical (cytokine/chemokine expression) variables are simultaneously analyzed. All data should be presented, regardless of significance or not. I also don't understand why early and late OSCC was grouped together in an analysis that aims to evaluate the probability of a cytokine/chemokine being associated with metastases.
2. Two major clinical variables that can impact cytokine expression profile and disease progression were missed here. HPV-status and alcohol consumption. These are also important etiological/risk factors that contribute to carcinogenesis and should be evaluated as well.
3. I don't understand the value of the pearson correlation analysis (Table 4) within the different cohort groups. Lines 241-243 state: "The complex relationship among these immune modulators in OSCC was demonstrated by Pearson’s correlation analysis displaying multiple significant positive interconnections between the investigated markers (Table 4)". Isn't this relationship expected and already established? Since they are NF-kB dependent?
4. The introduction lacks any information on the risk factors and/or etiological factors that contributes to these cancerous/pre-cancerous pathologies. Also, does not tell us about any differences between HL or PVL, same for early vs advanced OSCC.
minor issues:
1. Many acronyms used before being defined. You have defined most in the abstract, yes. But it needs to be defined in the introduction as well.
2. Many inconsistencies in grammar and spelling throughout the manuscript that need to be addressed. A lot of word choices/uses did not come across as the author's intended.
3. Should be "sex" and not "gender". The difference: sex is biological, whereas gender is social.
4. Don't understand the use of "Stage 1" in the title
Author Response
English language editing has been performed with revisions being traceable using the "Track Changes" function in the Microsoft Word format of the manuscript.
Major issues:
- Table 5 presents the results from testing the Bayesian hypothesis model which can be performed as an alternative to the standard univariate/ multivariate analysis and we considered as more intriguing according to our study objectives. It has been applied to predict the ​​critical level of quantification (LOQ) of each cytokine so that one or more analytes are determinants as a marker or risk markers related to a pathology group of interest, given the presence or not of other specific clinical phenotypes. The applied method involved the analysis of all clinical (presented in Table 1) and biochemical (cytokine/chemokine mean salivary levels ) variables showing conditional probabilistic relationships among them. The probability of association between altered cytokine levels and patients' clinical parameters is expressed as P values and considered as notable when P is greater than 0.5. In our study, according to the analysis shown in Table 5, the probability of association between TNF-α (at a concentration of about 155-157 pg/mL) and IL-6 (319-328 pg/ml) and advanced OSCC is 0.77 or 77% and 0.76 or 76%, respectively. In addition, the probability of association between IL-6 (308-317 pg/mL) and the presence of neck metastasis is 0.81 or 81%. To estimate an association of cytokine/chemokine modulation and the presence of neck metastases we joined the two OSCC cohorts to provide higher n for statistical analysis, as this parameter was present in only 6 patients from the early OSCC group (per Table 1).
- Information on the HPV-status and alcohol consumption habits was not available for all the recruited study participants according to their clinical history records. In addition, no correlation has been found between HPV infection and the etiology of OPMDs such as OL (García-López R, Moya A, Bagan JV, Pérez-Brocal V. Retrospective case-control study of viral pathogen screening in proliferative verrucous leukoplakia lesions. Clin Otolaryngol. 2014 Oct;39(5):272-80. doi: 10.1111/coa.12291. PMID: 25099922.). Alcohol has not been associated to the etiology of oral leukoplakia (Warnakulasuriya S, Kujan O, Aguirre-Urizar JM, Bagan JV, González-Moles MÁ, Kerr AR, Lodi G, Mello FW, Monteiro L, Ogden GR, Sloan P, Johnson NW. Oral potentially malignant disorders: A consensus report from an international seminar on nomenclature and classification, convened by the WHO Collaborating Centre for Oral Cancer. Oral Dis. 2020 Oct 31. doi: 10.1111/odi.13704. Epub ahead of print. PMID: 33128420.).
- Chemokines are secreted in response to signals such as proinflammatory cytokines like interleukin (IL)-1, tumor necrosis factor (TNF), and interferon-c (IFN-c). Pearson’s correlation analysis was carried out to estimate associations between the target cytokines as in normal healthy conditions, as in OSCC. Most of the significantly altered cytokines according to the non-parametric pair-wise comparisons showed significant positive correlations which supports the previous findings. However, no correlation was observed among IL-6 and MCP-1 expression levels in early and advanced OSCC, as it could have been expected according to DOI: 10.1016/s1074-7613(00)80334-9
- Information regarding the major risk factors that contribute to OSCC (lines:35-39), as well as the difference between HL and PVL (lines: 46-49) was added into the Introduction section of the manuscript.
Minor issues:
- Acronyms defined in the Abstract were also defined in the Introduction section, as recommended (lines: 31, 45, 47, etc.).
- 2. Grammar and spelling errors throughout the manuscript were revised (traceable using the "Track Changes" function in MS Word).
- "gender" was substituted with "sex" (line:146,330 and Table 1).
- Stage 1 indicates initial manuscript submission and review. It is not part of the title of the article and will be removed in the further processing of the manuscript.
Round 2
Reviewer 1 Report
Thank you for your speedy corrections. There remain English grammar and most importantly punctuation issues especially with the yellow highlighted additions. Please take care to remedy these in your next revision.
Thank you for adding the statement about the periodontal status of the patients and also provide feedback to the reviewer comments. More detail is required in your manuscript to allay any fear that patients with chronic oral conditions were included in tour study.
The issue regarding the medical status of the patients is NOT well considered and not satisfactory as an answer. The quoted reference discussing terminology of OPMD and your answer specifically made mention of the lack of an association between underlying medical problems and OSCC or OL. This reviewer would disagree that there are no systemic conditions which do not predispose to OL or OSCC formation. New literature is available on this topic. Nonetheless, that is not the main concern with my original statement. I did not mention a specific association in relation to etiology. My main concern is that patient with underlying systemic conditions will exhibit circulating cytokines, and these have been detected in saliva. If you make the assumption that all the salivary cytokines found in saliva only originate or are related to the oral tissues, then that is an incorrect assumption. It is well known that systemic cytokines can be detected in saliva, as can circulating DNA for example. Because of this, knowledge of the medical status of the patients is paramount. The readers need to know if the patients had an underlying immune conditions or inflammatory condition or another condition which could have resulted in systemic cytokine production that may have been picked up in the saliva. You must explicitly state that for all patients. If you did not collect this information (that would be unusual given what you have articulated in the manuscript), then you need to add further text to the limitations of this study to highlight these deficiencies. This must occur despite the lack of associations noted in your reply, as the methodology for salivary cytokine biomarkers in the literature has suffered form these problems, and adding another paper to the literature which does not satisfactorily address this problem will only weaken the usefulness of your results. The other way to explain this request is that if a patient displayed this cytokine profile but instead had liver disease and no evidence of oral cancer, would you still consider your cytokine profile an accurate one for presence of oral cancer?
Lastly, Erythroplakic (not Erithroplakic) is still incorrectly spelt in Table 1. Should be y not i.
Author Response
Dear Editors and reviewers,
We are grateful for your effort and helpful comments received, and for the opportunity to improve the quality and scientific level of this research. The reviewers’ suggestions made to the present study have been addressed in our careful revision. The responses to the comments are listed below.
Reviewer report:
Thank you for your speedy corrections. There remain English grammar and most importantly punctuation issues especially with the yellow highlighted additions. Please take care to remedy these in your next revision.
Thank you for adding the statement about the periodontal status of the patients and also provide feedback to the reviewer comments. More detail is required in your manuscript to allay any fear that patients with chronic oral conditions were included in tour study.
The issue regarding the medical status of the patients is NOT well considered and not satisfactory as an answer. The quoted reference discussing terminology of OPMD and your answer specifically made mention of the lack of an association between underlying medical problems and OSCC or OL. This reviewer would disagree that there are no systemic conditions which do not predispose to OL or OSCC formation. New literature is available on this topic. Nonetheless, that is not the main concern with my original statement. I did not mention a specific association in relation to etiology. My main concern is that patient with underlying systemic conditions will exhibit circulating cytokines, and these have been detected in saliva. If you make the assumption that all the salivary cytokines found in saliva only originate or are related to the oral tissues, then that is an incorrect assumption. It is well known that systemic cytokines can be detected in saliva, as can circulating DNA for example. Because of this, knowledge of the medical status of the patients is paramount. The readers need to know if the patients had an underlying immune conditions or inflammatory condition or another condition which could have resulted in systemic cytokine production that may have been picked up in the saliva. You must explicitly state that for all patients. If you did not collect this information (that would be unusual given what you have articulated in the manuscript), then you need to add further text to the limitations of this study to highlight these deficiencies. This must occur despite the lack of associations noted in your reply, as the methodology for salivary cytokine biomarkers in the literature has suffered form these problems, and adding another paper to the literature which does not satisfactorily address this problem will only weaken the usefulness of your results. The other way to explain this request is that if a patient displayed this cytokine profile but instead had liver disease and no evidence of oral cancer, would you still consider your cytokine profile an accurate one for presence of oral cancer?
Lastly, Erythroplakic (not Erithroplakic) is still incorrectly spelt in Table 1. Should be y not i.
Answer to the reviewer:
We agree with the reviewer because it is true that the underlying systemic conditions of the patients may influence the levels of interleukins in saliva. The literature describes several autoimmune conditions and infectious diseases that may alter them. In this sense, regarding periodontal disease, under Material and Methods we have added that oral health and periodontal status were recorded in all the volunteers recruited in the different groups, and those presenting significant oral periodontal disease were not considered for analysis.
There are some interesting articles showing what the reviewer mentions, as detailed below:
-Nibali L, Fedele S, D'Aiuto F, Donos N. Interleukin-6 in oral diseases: a review. Oral Dis. 2012 Apr;18(3):236-43. doi: 10.1111/j.1601-0825.2011.01867.x. Epub 2011 Nov 4. PMID: 22050374.
-Silvestre-Rangil J, Bagán L, Silvestre FJ, Martinez-Herrera M, Bagán J. Periodontal, salivary and IL-6 status in rheumatoid arthritis patients. A cross-sectional study. Med Oral Patol Oral Cir Bucal. 2017 Sep 1;22(5):e595-e600. doi: 10.4317/medoral.21937. PMID: 28809379; PMCID: PMC5694182.
-Schmidlin PR, Dehghannejad M, Fakheran O. Interleukin-35 pathobiology in periodontal disease: a systematic scoping review. BMC Oral Health. 2021 Mar 20;21(1):139. doi: 10.1186/s12903-021-01515-1. PMID: 33743678; PMCID: PMC7981974.
-Mozaffari HR, Sharifi R, Sadeghi M. Interleukin-6 levels in the serum and saliva of patients with oral lichen planus compared with healthy controls: a meta-analysis study. Cent Eur J Immunol. 2018;43(1):103-108. doi: 10.5114/ceji.2018.74880. Epub 2018 Mar 30. PMID: 29731693; PMCID: PMC5927179.
-Tishler M, Yaron I, Shirazi I, Yaron M. Saliva: an additional diagnostic tool in Sjögren's syndrome. Semin Arthritis Rheum. 1997 Dec;27(3):173-9. doi: 10.1016/s0049-0172(97)80017-0. PMID: 9431589.
For example, a very interesting article by Nibali et al. (2012) described that some systemic diseases, such as Sjögren’s syndrome (autoimmune disease), and other immune-mediated disorders such as oral lichen planus present alterations in salivary interleukins levels.
However, none of our cases presented such autoimmune or immune-related diseases that could have influenced the levels recorded in our results.
We have also published an article (Silvestre-Rangil et al., 2017) on another autoimmune disease (rheumatoid arthritis), in which there were also alterations in IL-6 levels in saliva. None of our cases presented rheumatoid arthritis, however.
Under Material and Methods we have added that we could not identify any other systemic diseases (such as autoimmune disorders) in our groups that may have influenced our salivary findings.
Some systemic conditions may predispose to the development of OL or OSCC according to the reviewer, such as for example anemia (Plummer Vinson's syndrome), terciary syphilis, oral lupus erythematous and oral graft-versus-host disease, but none of our cases presented any of these disorders.
Finally, we have modified Erythroplakic in Table I.
Reviewer 3 Report
Regarding Authors Rebuttal # 1:
Okay, the reasoning for this choice in using the Bayesian hypothesis model is valid.
Regarding Authors Rebuttal # 2:
Yes, I do understand that HPV infection is not involved in the etiology of OPMDs. However, it is definitely involved in the etiology of oral cancers, as is excessive alcohol consumption. So the oral cancer samples you used could very well be HPV+, since there is no information available on their HPV-status. This is a confounding variable, that in my opinion, needs to be stated in the materials and methods, to make the reader aware that these samples lack that vital information.